# New Copper(I) Complex with a Coumarin as Ligand with Antibacterial Activity against *Flavobacterium psychrophilum*

**DOI:** 10.3390/molecules25143183

**Published:** 2020-07-13

**Authors:** Maialen Aldabaldetrecu, Mick Parra, Sarita Soto, Pablo Arce, Mario Tello, Juan Guerrero, Brenda Modak

**Affiliations:** 1Laboratory of Coordination Compounds and Supramolecularity, Faculty of Chemistry and Biology, University of Santiago of Chile, 9170002 Santiago, Chile; maialen.aldabaldetrecu@usach.cl (M.A.); pablo.arce@usach.cl (P.A.); 2Laboratory of Natural Products Chemistry, Centre of Aquatic Biotechnology, Faculty of Chemistry and Biology, University of Santiago of Chile, 9170002 Santiago, Chile; mick.parra@usach.cl; 3Laboratory of Bacterial Metagenomic, Centre of Aquatic Biotechnology, Faculty of Chemistry and Biology, University of Santiago of Chile, 9170002 Santiago, Chile; mario.tello@usach.cl; 4Laboratory of Biochemistry and Oral Biology, Faculty of Odontology, University of Chile, 8380000 Santiago, Chile; sarita.soto@usach.cl

**Keywords:** coumarin, copper (I), coordination compounds, *F. psychrophilum*, antibacterial activity

## Abstract

A new copper (I) complex, [Cu(NN_1_)_2_](ClO_4_)***_,_*** was synthesized, where NN_1_ was a imine ligand 6-((quinolin-2-ylmethylene)amino)-2H-chromen-2-one obtained by a derivatization of natural compound coumarin. The structural characterization in solution was done by NMR techniques, UV-Vis and cyclic voltammetry. The potential antibacterial effect of [Cu(NN_1_)_2_](ClO_4_)***_,_*** was assessed for *F. psychrophilum* isolated 10094. *F. psychrophilum* is a Gram-negative bacterium which causes diseases such as bacterial cold-water disease and rainbow trout fry syndrome, causing large economic losses in the freshwater salmonid aquaculture industry. This complex show to have antibacterial activity against *F. psychrophilum* 10094 at non-cytotoxic concentration in cell line derived from trout (*F. psychrophilum* 10094 IC_50_ 16.0 ± 0.9; RT-GUT IC_50_ 53.0 ± 3.1 µg/mL).

## 1. Introduction

*Flavobacterium psychrophilum* is a Gram-negative bacterium, causative of septicemic disease “cold-water disease” (CWD) or “rainbow trout fry syndrome” (RTFS) in freshwater fish species worldwide and increased susceptibility to other diseases [1] generating serious economic losses. It is also present in non-salmonid fish as *Anguilla anguilla, Plecoglossus altivelis* and *Tinca tinca* [2]. In Chile, the second largest farmed salmon producer in the world after Norway, *Flavobacterium psychrophilum* is the fourth pathogen that produces more mortalities in Rainbow trout, after *Piscirickettsia salmonis*, IPNV and *Vibrio*, considered within high risk diseases [3]. The control and prophylactic strategies are based mainly on the use of antibiotics and vaccination. The antibiotics used in Chile are mainly florfenicol and oxytetracycline [4,5]. However, the indiscriminate antibiotic use produces alterations of the bacteria flora in the aquatic environment inducing the appearance of resistant bacteria [4], and potentially transfer resistance to pathogens terrestrial animals and humans [6]. Identifying novel antibacterial drugs is therefore of critical importance and natural products are an excellent source for such discoveries. The plant kingdom constitutes a source of new chemicals, which may be important for their potential use as antibacterial. One natural compound that has been studied for its antibacterial properties is coumarin. Coumarin is basically made up of a benzene moiety fused with an alpha-pyrone ring named as benzopyrone [7]. Various naturally isolated coumarins, as well as their chemically modified analogs, are active against numerous bacterial strains, including those which have developed multidrug resistance [7]. Coumarin (1-benzopyran-2-one) itself has a low antibacterial activity, but its derivatives as aegelinol and agasyllin have shown significant antibacterial activity against clinically isolated Gram-positive and Gram-negative bacterial strains such as *Staphylococcus aureus*, *Salmonella typhi*, *Enterobacter cloacae* and *Enterobacter aerogenes* [7].

On the other hand, in the search for new treatments and as an alternative to antibiotics, the antibacterial properties of various metals, including copper, have been studied in recent years. Several researchers have reported the antibacterial effect of copper compounds in bacteria, such as *Salmonella enterica*, *Campylobacter jejuni* and *Escherichia coli* [7,8,9].

There are several mechanisms of action described for the both Cu(I) and Cu(II) complexes, one of them is the oxidative stress. Few works of antibacterial activity of Cu(I) was published, however the redox activity of Cu(I) makes it a powerful catalyst of Fenton and Haber-Weiss type reaction, which lead to the generation of the extremely reactive hydroxyl radical and less dangerous reactive oxygen species (ROS). The hydroxyl radical can react with a wide range of biologic molecules, including DNA, causing cellular irreversible damage [10,11].

Based on this background, our target was to develop a new compound, based on a product such as coumarin, which together with copper, formed a complex with good antibacterial activity with nontoxic concentrations of copper. In this study, we report the synthesis of a new Cu(I) coordination complex [Cu(NN_1_)_2_]ClO_4,_ (Figure 1) where NN_1_ = 6-((quinolin-2-ylmethylene)amino)-2H-chromen-2-one, a ligand derivate from coumarin 1-benzopyran-2-one, which presented a remarkable antibacterial activity against *Flavobacterium psychrophilum*, at non-cytotoxic concentrations.

## 2. Results and Discussion

### 2.1. Synthesis of Cu(I) Coordination Compound [Cu(NN_1_)_2_]ClO_4_, Where NN_1_ Is a Ligand Obtained from Coumarin 1-benzopyran-2-one

A new imine ligand 6-((quinolin-2-ylmethylene)amine)-2*H*-chromen-2-one (NN_1_) derivate from coumarin and their homoleptic Cu(I) complex of formula [Cu(NN_1_)_2_]ClO_4_ were prepared in good yield and high purity.

The NN_1_ ligand was obtained by several steps as showed in the scheme of Figure 1, that involved nitration of coumarin by knowledge nitration method [12], and its subsequent reduction to 6-aminocoumarin with Fe powder [13]. The condensation reaction of this amine with 2-quinoline-carboxaldehyde by microwave assisted reaction yield the imine ligand NN_1_, as a straw yellow crystalline compound (Figure 1) [14].

Finally, the reaction of two equivalents of NN_1_, with an equivalent of [Cu(CH_3_CN)_4_]ClO_4_ precursor complex under mild reaction conditions produces a purple solution from which the complex were obtained in high purity as microcrystalline power [15,16,17].

### 2.2. Structural Characterization NMR

^1^H NMR spectra supported the structure of all intermediate products (Figure 2 and Appendix A). An unequivocal structure of NN_1_ and complex [Cu(NN_1_)_2_]ClO_4_ were established by a concerted use of 1D, ^1^H, {^1^H}^13^C NMR and 2D COSY, ^1^H ^13^C HSQC, ^1^H ^13^C HMBC and ^1^H ^15^N HMBC (Appendix A). The unequivocal signals assignment (^1^H and ^13^C) are reported in experimental.

The unambiguous proton and carbon signals assignment of complex and ligand was performed as follows: all protons and carbons bonded together were identified by HSQC spectra. The singlet proton signal placed around to 9.00 ppm, assigned to imine proton was corroborated for ^1^H ^15^N HMBC and ^1^H ^13^C HMBC spectra which also confirming the formation of imine group linking the quinoline and coumarin fragments as shown in Figure 2. The discrimination between the different protons spin system for each molecular fragment was realized by concerted analysis of ^1^H ^13^C HMBC and COSY spectra. The quaternary carbon and relative position of oxygens atom was established by the ^1^H ^13^C HMBC technique. Characteristic proton and carbon spectra were observed for free ligand recorded in CDCl_3_ with expected frequency ranges for this type of organic molecules. On the contrary, even considering the impact of the solvent, the ligand signals in the complex show the effect of metal coordination which shifts the imine proton to high frequencies. We can presume that this effect is expected by the extraction of electronic density from nitrogen by the metal complexation. However, copper (I) is a weak Lewis acid to explain this high deshielding by itself, but can synergistically act with magnetic currents of the aromatic ring of the ligand that are placed orthogonally to each other to increase the deshielding effect on imine proton and explain shielding effect on H8 (5.94 ppm).

### 2.3. Characterization in Solution

#### 2.3.1. UV-Vis

Electronic spectra data of the complexes and ligands in CHCl_3_ are shown in Figure 3. The spectral region below 400 nm in complex correspond at π to π* intraligand transition bands, which are similar profile of free ligand, but undergo a red-shift upon complexation to copper(I) in around 20 nm.

The broadband centered in 562 nm for the [Cu(NN_1_)_2_]^+^ complex correspond at one metal to ligand charge transfer band (MLCT) (Figure 3) in accordance with their intensity and energy. In addition, this band showed a similar spectral profile to shown by homoleptic [Cu(biq)_2_]^+^ complex (λ-max TCML is 550 nm) which biq ligand (where biq correspond to 2,2′-biquinoline) has structural similarities with NN_1_ [18]. Like in that complex, this MLCT is a transition of HOMO mainly centered in Cu(I) with full d^10^ shell, toward LUMO formed by π* orbital of the 2-imine-quinoline fragment of NN_1_ ligand. Another band observed as a shoulder around to 416 nm, can correspond to a second MLCT transition from metal to π* orbital of coumarin fragment of ligand. The assignment of second MLCT is supported by its most blue-shifted position in accordance with the fact that the π* orbital of coumarin is of greater energy than π* orbital of the 2-imine-quinoline fragment, appearing overlapped by the more intense intraligand bands. 

No significant changes in the spectral bands were observed at 72 h after the solution was prepared showing high stability in that solvent and air-exposed at room temperature.

#### 2.3.2. Cyclic Voltammetry

Voltammetry studies were also performed in order to determine the effect of new NN_1_ ligand on the Cu(II)/Cu(I) oxidation potential considering the well-known fact of high influence of ligands on the metal-centered HOMO energy. Consequently, measurements were registered at room temperature in −1.10 to +1.30 V vs Ag/AgCl potential range. Figure 4 shows representative cyclic voltammograms for ligand (NN_1_) and respective coordination complex [Cu(NN_1_)_2_](ClO_4_) in dichloromethane.

The free ligand undergoes two irreversible oxidation waves within the potential window allowed by the solvent at potential greater than 0.9 V. On the contrary, the complex [Cu(NN_1_)_2_]ClO_4_, shows only one quasireversible oxidation process assignable to the Cu(II)/Cu(I) couple in E_1/2_ = 0.426 V (ΔE = 114 mV)]. Scan rate dependence reveals that the copper complexes are oxidized in a diffusion-controlled step (Appendix A).

The oxidation potentials occur at less positive potentials compare well to these reported for similar, [Cu(biq)_2_]ClO_4_ complex which exhibits a value E_1/2_ = 0.96 V versus Ag/AgCl in CH_2_Cl_2_ [18]. The lower oxidation potential for the new Cu(I) complex [Cu(NN_1_)_2_]ClO_4_, was associated with distortion of the ideal geometry (tetrahedral) for the metal center toward a more flattened structure which is associated with increased of its reactivity [19] which is an interesting property to designed biologically active drugs.

### 2.4. Antibacterial Activity against Flavobacterium psychrophilum

#### 2.4.1. Antibacterial Test

The antibacterial activity of [Cu(NN_1_)_2_]ClO_4_ complex against *F. psychrophilum* 10094, [Cu(CH_3_CN)_4_]ClO_4_ salt and coumarin was evaluated, for which Minimum Inhibitory Concentration MIC, Minimum Bactericidal Concentration MBC and Half Maximal Inhibitory Concentration IC_50_ of these compounds were determined using concentrations between 512-μg/mL to 2 μg/mL. The results are shown in Table 1. MIC and MBC values do not show standard deviations since each replicate separately showed a similar behavior according to the evaluated treatment (View experimental section).The IC_50_ of the new copper (I) complex was 16.1 ± 0.9 µg/mL, slightly higher in comparison with the IC50 of the [Cu(CH_3_CN)_4_]ClO_4_ (10.4 ± 0.7). However, the concentration of [Cu(NN_1_)_2_]ClO_4_ complex necessary to inhibit growth and kill the bacteria (MIC and MBC of 32 µg/mL) was lower than the observed by the precursor complex (MIC and MBC of 64 µg/mL). The effect of coumarin on the *F. psychrophilum* 10094 was only observed at high concentration of the compound (IC_50_ 160.0 ± 25.5 µg/mL, MIC 512 µg/mL and MBC < 512 µg/mL), despite the fact that the antibacterial effect of coumarin in a large number of bacteria has been reported [20]. Thus, the copper (I) complex shown an antibacterial effect more similar to the precursor complex, but with higher potency. Altogether, these data suggest that in the copper (I) complex, the metallic center Cu(I) is the main responsible for the antibacterial activity against *F. psychrophilum* 10094.

In this sense, the increase in antibacterial capacity observed by the copper (I) complex, [Cu(NN_1_)_2_]ClO_4_, could be given because the coordination of the metal ion with NN_1_ ligand which would increase the cellular permeability increasing the amount of copper inside the cell [21]. The above occur since the metal polarity is reduced by chelation of copper (I) complex is with NN_1_ ligand, increasing the delocalization of π-electrons and enhances the lipophilicity of the complex. This increased lipophilicity in turn enhances the penetration of the complex into lipid membranes and blocking of metal binding sites on the enzymes of the microorganisms [22].

Although the antibacterial mechanism of this complex was not addressed in this work, it is known that copper ions are available to produce Fenton-type reactions, generating hydroxyl radicals, which could induce oxidative stress, either at the cytoplasm or inner membrane, which could kill the bacteria [23,24].

Because a high concentration of copper can be toxic, we determine the amount of copper present in the copper (I) complex and in the precursor of the complex. As show in Table 2 the IC_50_, MIC as well as in the MBC of the precursor salt have 2.3 times, respectively of amount of copper in relation to the complex with the NN_1_ ligand, that is to say, the percentage of copper present in the precursor is 19.5%, while in the complex is only 8.3%. These results demonstrate the importance of the presence of the NN_1_ ligand in the antibacterial activity decreasing the amount of copper needed and therefore toxicity.

#### 2.4.2. Cytotoxicity Test

Since *F. psychrophilum* is a bacterial pathogen that affects a wide variety of salmonids, we evaluated the toxicity of the [Cu(NN_1_)_2_]ClO_4_ complex in three different cell lines, Chinook salmon embryo (CHSE-214), Salmon Head Kidney-1 (SHK-1) and intestinal epithelial cell line (RT-GUT). For this experiment 1 × 10^5^ cells were treated with the different compounds (coumarin, Cu(I) salt and copper (I) complex) in a range of concentrations between 512 μg/mL and 2 μg/mL, incubated at 16 °C for 24 h. The cellular viability was analyzed by flow cytometry and the results are shown in Table 1. In the copper (I) complex and precursor complex [Cu(CH_3_CN)_4_]ClO_4_, a concentration-dependent toxicity was observed. On the other hand, the IC_50_ of each compound was different depending on the cell line used, it was observed that in the case of the copper (I) complex the compound is more toxic in cell lines derived from salmon, with an IC_50_ of 29.1 ± 1.4 µg/mL, while it is less toxic in a line derived from trout with an IC_50_ of 53.0 ± 3.1 µg/mL in RT-GUT. Similar effect is observed with the precursor complex, where the compound is more toxic in the CHSE-214 cell line with an IC_50_ of 59.4 ± 4.1 µg/mL, being less toxic in the RT-GUT cell line with an IC_50_ of 233.9 ± 19.5 µg/mL. This difference in the effect of copper administration is also observed in vivo when different salmonid species are fed with CuSO_4_, where the adverse effects depend on the species and growth stage [25]. Finally, coumarin toxicity was not observed in any of the cell lines used. The increase in the toxicity of the copper (I) complex compared to coumarin, was probably due to the ability of coumarin to permeate the membrane [21], functioning as a carrier for the copper metal center of the complex. On the other hand, analyzing the effect of the copper (I) complex, [Cu(NN_1_)_2_]ClO_4,_ on the different cell lines, our results shows that the NN_1_ ligand increase the toxicity of [Cu(NN_1_)_2_]ClO_4_, more in eukaryotic cells than for prokaryotic cells. This effect is interesting, since it is necessary to carry out experiments to elucidate the mechanism of action of [Cu(NN_1_)_2_]ClO_4_ in both prokaryotic and eukaryotic cells and how the cell wall of prokaryotic cells influences the toxicity of [Cu(NN_1_)_2_]ClO_4_. In addition, although a higher toxicity of [Cu(NN_1_)_2_]ClO_4_ is observed in all cell lines, compared to the copper salt, [Cu(CH_3_CN)_4_]ClO_4,_ more promising results could be obtained in fish. The stability to stomach and intestinal pH, as well as capacity to be absorbed in the gut, are variable not present in cultures conditions producing that in general in vivo models show greater resistance than in vitro studies [26,27,28]. Interestingly, the results showed that the RT-GUT cell line was more resistant to [Cu(NN_1_)_2_]ClO_4_, suggesting that this compound could be applied at the mucosa level in rainbow trout either in baths or through food, although still remains to determine if this compound have toxic effects on the fish to the concentration assessed with *F. psychrophilum*.

## 3. Experimental Section

### 3.1. General Methods

All the chemicals were purchased commercially and used without further purification. ^1^H, ^13^C NMR and 2D spectra were recorded on a Bruker Avance Neo 400 MHz spectrometer (400.133 MHz for ^1^H, 40.559 MHz for ^15^N and 100.613 MHz for ^13^C, (University of Santiago of Chile, Santiago, RM, Chile) equipped with a 5 mm multinuclear broad-band dual probehead, incorporating a z-gradient coil. All the measurement was done in CDCl_3_ at 300 K. Chemical shifts (in ppm) for ^1^H and ^13^C were calibrated respect to the residual protonated signal of the solvent (7.26 and 77 ppm respectively) and reported relative to Me_4_Si. The ^15^N signal in ^1^H ^15^N HMBC were calibrated relative to CH_3_NO_2_ (90%) in CDCl_3_). The following abbreviations were used to explain the multiplicities: s = singlet, d = doublet, m = multiplet. Electronic spectra were recorded on a PerkinElmer UV-VIS Lambda 25 spectrophotometer (University of Santiago of Chile, Santiago, RM, Chile) at room temperature, wavelength range between 200 and 1100 nm from 1.5 × 10^−5^ mol/L solutions in dichloromethane to NN_1_ and [Cu(NN_1_)_2_]ClO_4_. Electrochemical behavior was studied by cyclic voltammetry using a potentiostat CHI 620E by CH instrument equipment supplied with the electrochemical analyzer CHI software (University of Santiago of Chile, Santiago, RM, Chile). A standard electrochemical three electrode cell by CH instruments, 10 mL volume provided with a glassy carbon working electrode, a platinum wire auxiliary electrode and a AgCl/Ag reference electrode. Measurements were performed at room temperature and N_2_ atmosphere in 2 mmol/L to NN_1_ and 1 mmol/L to [Cu(NN_1_)_2_]ClO_4_ in dichloromethane solutions using tetrabutylammonium perchlorate (c.a. 0.1 mol/L) as supporting electrolyte, at several scan rates (50 to 300 mVs^−1^ range). The potentials were informed as E_1/2_
*v*/*s* Ag/AgCl.

### 3.2. 6-Nitrocoumarin

Coumarin was nitrated with mixed acid in an ice bath. Coumarin (1.00 g, 7.1 mmol) was dissolved in conc. H_2_SO_4_ (5 mL) and temperature was maintained at 0 °C and then 5 mL mixed acid (HNO_3_ and H_2_SO_4_ (conc.) in 1:3 volume ratio) was added. The mixture was stirred keeping at 0 °C for 1 h. Water at 0 °C (50 mL) was added to precipitate obtaining A white precipitate of 6-nitrocoumarin. It was then filtered and washed thorough with cold water (10 mL). The compounds were dried a 60 °C overnight. The identity of the compound was determined by ^1^H NMR and used as was obtained. Yield, 1.20 g (88%). ^1^H NMR (400.133 MHz, CDCl_3_) δ/ppm H5′ 8.43 (d, ^4^*J*_5′–7′_ = 2.4 Hz, 1H), H7′ 8.40 (dd, ^3^*J*_7’–8′_ = 9.0, ^4^*J*_7′–5′_ = 2.5 Hz, 1H), H4′ 7.80 (d, ^3^*J*_4′–3′_ = 9.7 Hz, 1H), H8′ 7.47 (d, ^3^*J*_8′–7′_ = 9.0 Hz, 1H), H3′ 6.59 (d, ^3^*J*_3′–4′_ = 9.7 Hz, 1H). (Appendix A).

### 3.3. 6-Aminocoumarin

Reduction of 6-nitrocoumarin was done using iron powder and ammonium chloride in water. 6-Nitrocoumarin (1.00 g, 5.2 mmol) in water (150 mL) was treated with Fe-powder (2.00 g, 36 mmol) and ammonium chloride (0.24 g, 4.5 mmol). The mixture was kept in water bath for 12 h with stirring. A dark brown precipitate was obtained which was then extracted with dichloromethane. 6-aminocoumarin was obtained as yellow precipitate by slowly dichloromethane evaporation finally washed by two times with few milliliters of dry diethyl ether, and the identity of the compound was determined by ^1^H NMR. Yield, 0.64 g (76%). ^1^H NMR (400.133 MHz, CDCl_3_) δ/ppm H4′ 7.57 (d, ^3^*J*_4′–3′_ = 9.5 Hz, 1H), H_8′_ 7.15 (d, ^3^*J*_8′–7′_ = 8.8 Hz, 1H), H7′ 6.87 (dd, ^3^*J*_7′–8′_ = 8.8, ^4^*J*_7′–5′_ = 2.7 Hz, 1H), H5′ 6.71 (d, ^4^*J*_5′–7′_ = 2.7 Hz, 1H), H3′ 6.38 (d, ^3^*J*_3′–4′_ = 9.5 Hz, 1H), NH_2_ 3.71 (s, 2H) (Appendix A).

### 3.4. Ligand: NN_1_

The synthesis was carried out by microwave reaction in a Teflon reactor. In it, 6-amino coumarin(N_1_) (0.50 g, 3.1 mmol) and 2-quinoline-aldehyde (N) (0.49 g, 3.1 mmol) of were added, solubilizing them in 5 mL of benzene. It was irradiated for 14 min obtaining a pale-yellow solid. It was washed with dichloromethane (30 mL) and ethyl ether (50 mL). It was recrystallized by solubilizing the powder in hot methanol, filtered on porous glass filter and slowly solvent evaporation at room temperature, obtained a pale-yellow microcrystal. Yield, 0.86 g, 92%. ^1^H NMR (400.133 MHz, CDCl_3_) δ/ppm Hi 8.82 (s, 1H), H3 8.35 (d, ^3^*J*_3–4_ = 8.6 Hz, 1H), H4 8.28 (d, ^3^*J*_7–8_ = 8.6 Hz, 1H), H8 8.17 (d, ^3^*J*_7–8_ = 8.5 Hz, 1H), H5 7.89 (d, ^3^*J*_5–6_ = 8.0 Hz, 1H), H7 7.79 (t, ^3^*J*_7–8_ = 8.5 Hz, ^3^*J*_10–11_ = 7.7 Hz, 1H), H4′ 7.75 (d, ^3^*J*_3′–4′_ = 9.6 Hz, 1H), H6 7.63 (t, ^3^*J*_5–6_ = 8.0 Hz, ^3^*J*_6–7_ = 7.7 Hz, 1H), H7′ 7.57 (dd, ^3^*J*_7′–8′_ = 8.7, ^4^*J*_5′–7′_ =2.4 Hz, 1H), H5′ 7.47 (d, ^4^*J*_5′–7′_ = 2.3 Hz, 1H), H7′ 7.41 (d, *J* = 8.8 Hz, 1H), H3′ 6.48 (d, ^3^*J*_3′–4′_ = 9.5 Hz, 1H). ^13^C NMR (100.613 MHz, CDCl_3_) δ/ppm 194.26 C2, 162.10 Ci, 160.90 C10′, 154.76 C2, 153.28 C9′, 148.36 C9, 147.67 C6′, 143.66 C4′, 137.16 C4, 130.59 C7, 130.18 C8, 129.48 C10, 127.41 C6, 128.32 C5, 125.45 C7′, 120.30 C5′, 118.88 C3, 118.19 C8′, 117.75 C3′ (Appendix A).

### 3.5. Coordination Complex Cu(I): [Cu(NN_1_)_2_]ClO_4_

[Cu(CH_3_CN)_4_]ClO_4_ (0.47 g, 1.42 mmol) was taken in a 25 mL double neck round bottom flask dissolved in CH_3_CN by magnetic stirring and under N_2_ atmosphere. Then, NN_1_ (0.86 g, 2.852 mmol) was added dissolved in CH_3_CN and the reaction was continued under N_2_ atmosphere and stirring for 1 h. A purple- black precipitate was collected by filtration and dried. The compound was washed with diethyl ether and recrystallized. Yield 0.82 g (76%); ^1^H NMR (400.133 MHz, CDCl_3_) δ/ppm Hi 9.72 (s, 1H), H3 8.61 (d, *J* = 8.3 Hz, 1H), H4 8.40 (d, *J* = 8.3 Hz, 1H), H5′ 8.08–8.00 (m, 1H), H8 7.96–7.90 (m, 1H), (H7′, H5) 7.78 (t, *J* = 7.0 Hz, 2H), (H7, H6, H4′) 7.67–7.51 (m, 3H), H3′ 7.07 (d, *J* = 8.8 Hz, 1H), H8′ 6.23 (d, *J* = 9.6 Hz, 1H).^13^C NMR: (100.613 MHz, CDCl_3_) δ/ppm 159.97 C2′, 159.26 C10′, 154.27 C9′, 151.25 CI, 145.61 C9, 143.65 C7′, 142.84 C10, 139.14 C4, 132.30 C6, 131.16 C6′, 129.70 C7, 128.51 C8, 127.73 C5, 127.04 C4′, 124.92 C3, 121.68 C5′, 119.94 C2, 118.24 C3′, 117.50 C8′ (Appendix A).

### 3.6. Bacterial Strain and Growth Conditions

A bacterial strain of *Flavobacterium psychrophilum* (ETECMA) originally isolated from Salmon Coho (*Oncorhynchus kisutch*) in 2008 in Chiloe, Chile, was used for antibacterial test. Previous experiments performed in the laboratory analyzed the 16S gene of this bacterium and showed an identity of 99.6% with *Flavobacterium psychrophilum* 10094. The strain was cultivated in TYES broth (pH = 7.2) at 15 °C for 48 h with stirring at 180 rpm.

### 3.7. Antibacterial Test

To determine MIC, MBC and IC_50_ of complex and its precursors, microdilution method was performed [29] with some modifications. *F. psychrophilum* 10094 (0.4 OD_600_ nm = 1 × 10^8^ CFU/mL, approximately) was inoculated in TYES broth pH 7.2 to 1% cell suspension in 96-well plates (Falcon) and treated with complex and its precursors in serial concentrations between 512 μg/mL to 2 μg/mL and incubated for 72 h at 15 °C with agitation at 180 rpm. The MIC was determined as the lowest compounds concentrations which inhibited bacterial grow by visual comparison with the negative control (TYES broth). To determine the MBC, an aliquot of 10 μL from MIC was growth on TYES agar plates, pH 7.2, at 20 °C for 96 h. The MBC was determined as the minimum concentration of the compounds at which the bacterial colony count was diminish to 99.9%. For determination of the half-maximal inhibitory concentration (IC_50_) of compounds on *F. psychrophilum*, the OD_600_ nm of each wells was measured using Nanoquant Infinite M200 Pro (TECAN, Grödig, Austria). For the correct calculation of the OD of each concentration of each compound, wells with TYES medium were used with each concentration without inoculating with *F. psychrophilum*, this control OD was subtracted obtaining the final OD. Data were analyzed using the GraphPad Prism program 5.0. The concentrations were transformed into log (10) and OD_600_ nm normalized in percentage, for the calculation of IC_50_ a nonlinear regression was performed.

### 3.8. Cytotoxicity Test

Three different cell lines were used to assess the toxicity that the compounds could have on fish: Chinook salmon embryo CHSE-214 (ATCC CRL 1681), Salmon Head Kidney-1 SHK-1 and intestinal epithelial cell line RT-GUT. 1 × 10^5^ Cells were seeded and incubated at 16 °C for 24 h. Later, the cells were treated with compounds in concentrations from 512 μg/mL to 2 μg/mL solubilized in dimethyl sulfoxide (DMSO, Merck, Darmstadt, Germany) and incubated at 16 °C for 24 h. Subsequently, the cells were collected, pelleted and resuspended in IF buffer (1X buffer and 2% fetal bovine serum) and 1 µL propidium iodide (PI, 1 mg/mL) was added. The cellular viability (negative to PI) was analyzed by flow cytometer using FACSCanto II Cytometer (BD biosciences, University of Santiago of Chile, Santiago, RM, Chile). For positive control, cells were incubated without compounds; for death control, cells were incubated with 50% ethanol; medium control cells were treated with medium containing 1% of DMSO. To calculate the IC_50_ of compounds in the cell lines, they were analyzed using GraphPad Prism program 5.0 (GraphPad Software, San Diego, CA, USA), the concentrations were transformed into log (10) and the number of events normalized in percentage, for the calculation of IC_50_ a nonlinear regression was performed.

### 3.9. Statistical Analysis

Statistical analysis was performed using the GraphPad prism 5 program, using a nonparametric Mann-Whitney *t*-test analysis with a *p* < 0.05. The analyses were carried out separately for bacteria and for each cell line used.

## 4. Conclusions

A new Copper (I) coordination complex [Cu(NN_1_)_2_]ClO_4_, where NN_1_ is 6-((quinolin-2-ylmethylene)amino)-2H-chromen-2-one—a ligand derivate from coumarin was synthesized, characterized and evaluated for their antibacterial activity against the pathogen *F. psychrophilum.*

The NN_1_ ligand was obtained by several steps by derivatization of coumarin, a natural product with antibacterial activity. The complex was obtained by coordination the ligand to metal in soft reaction condition, with good yield. Its structure was unequivocally established by several NMR techniques, UV-Vis and Cyclic Voltammetry.

In vitro experiments carried out against *F. psychrophilum* 10094 showed that [Cu(NN_1_)_2_]ClO_4_ increased its antibacterial capacity compared to coumarin and precursor copper salt ([Cu(ACN_4_)]ClO_4_). Although the action mechanism of the copper (I) complex was not studied, the role of this metal ion in the antibacterial activity is evident. The complexation with NN_1_ ligand would oxidation state maintain and the antibacterial activity in lower amounts of copper making this type of complexes good candidates for the generation of new treatments against *F. psychrophilum*.

On the other hand, it was observed that in cell lines derived from rainbow trout mucosa (RT-GUT), the application of [Cu(NN_1_)_2_]ClO_4_ showed a more resistance to toxicity. However, is necessary to complement this with in vivo experiments to determine the real tolerance to complex (I) copper. In this way, the [Cu(NN_1_)_2_]ClO_4_ could be administered as therapy against *F. psychrophilum* through the fish mucosa by bath or through food. Furthermore, due to the difference in the cytotoxic effects observed of [Cu(NN_1_)_2_]ClO_4_ on the bacterial cell and eukaryotic cells (SHK-1, CHSE-214), it would be interesting in future studies to analyze the mechanisms by which [Cu(NN_1_)_2_]ClO_4_ exerts an effect on the permeability of the prokaryotic and eukaryotic cell membrane. With antibacterial activity assays, it is possible to assume that the administration of copper(I)-based drugs as the coordination complexes, is a safe and adequate antimicrobial strategy.

## Figures and Tables

**Figure 1 molecules-25-03183-f001:**
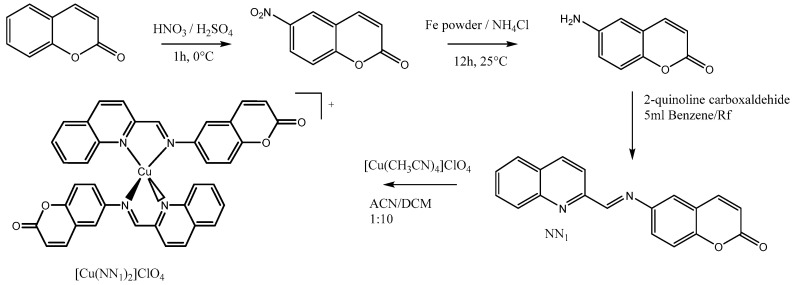
Synthesis scheme [Cu(NN_1_)_2_]ClO_4_.

**Figure 2 molecules-25-03183-f002:**
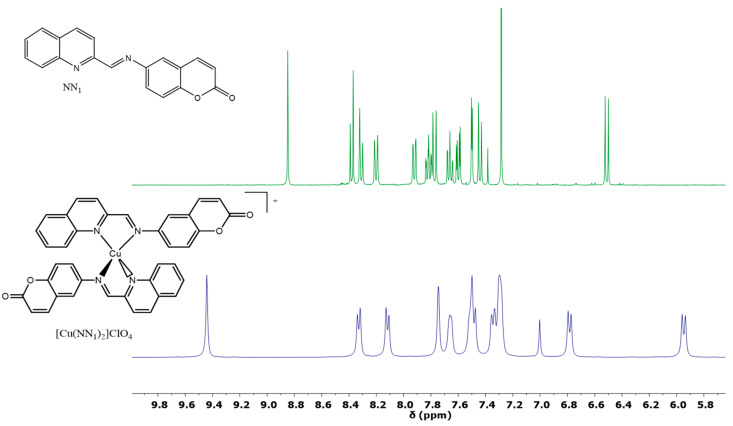
^1^H-RMN in CDCl_3_, NN_1_ (green) and [Cu^I^(NN_1_)_2_](ClO_4_) (blue) in CDCl_3_.

**Figure 3 molecules-25-03183-f003:**
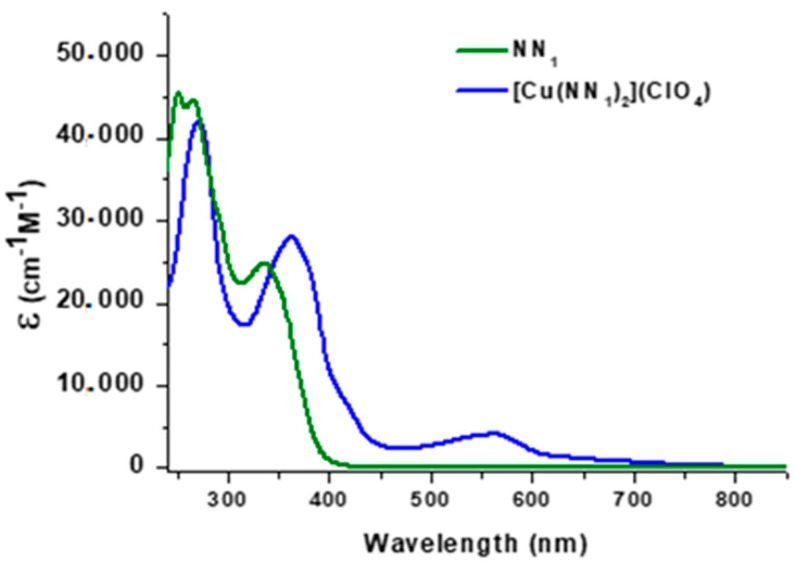
UV-Vis spectra for NN_1_ ligand (green) and complex [Cu(NN_1_)_2_]ClO_4_ (blue).

**Figure 4 molecules-25-03183-f004:**
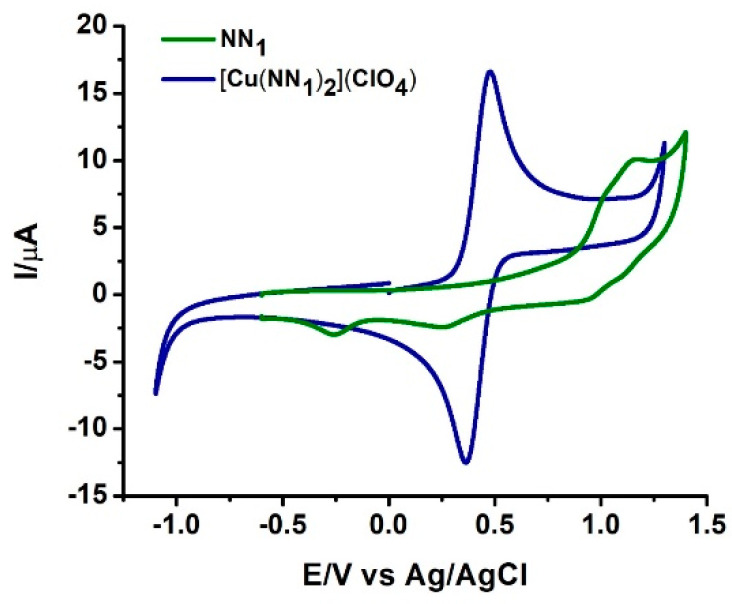
Characteristic cyclic voltammograms for ligand NN_1_ (green) and coordination complex [Cu(NN_1_)_2_](ClO_4_) (blue). NN_1_ and [Cu(NN_1_)_2_](ClO_4_) in (CH_2_Cl_2_) and 0.1 M TBAP. Potential are related to Ag/AgCl at 100 mV/S scan rate.

**Table 1 molecules-25-03183-t001:** Comparative table of MIC, MBC and IC_50_ on *F. psychrophilum* (*n* = 3) at 72 h post-incubation and IC_50_ on Chinook salmon embryo (CHSE-214), Salmon Head Kidney-1 (SHK-1) and intestinal epithelial cell line (RT-GUT) cells (*n* = 6; 3 independent replicas) of the copper (I) complex [Cu(NN_1_)_2_]ClO_4_. The precursors Coumarin and Cu(I) salt ([Cu(CH_3_CN)_4_]ClO_4_) were used like control.

*Flavobacterium psychrophilum* 10094				CHSE-214	SHK-1	RT-GUT
Compounds	IC_50_ (bacterial) µg/mL	MIC µg/mL	MBC µg/mL	IC_50_ (cellular) µg/mL	IC_50_ (cellular) µg/mL	IC_50_ (cellular) µg/mL
Coumarin	160.0 ± 25.5^**a**^	512	>512	>512	>512	>512
[Cu(CH_3_CN)_4_]ClO_4_	10.4 ± 0.7 **^b^**	64	64	59.4 ± 4.1 **^a^**	159 ± 44.6 **^a^**	233.9 ± 19.5 **^a^**
[Cu(NN_1_)_2_]*ClO*_4_	16.1 ± 0.9 **^c^**	32	32	29.1 ± 1.4 **^b^**	30.8 ± 1.3 **^b^**	53.0 ± 3.1 **^b^**

Statistical analysis was performed independently for the bacteria and for each cell line tested where different letters (**^a^**,**^b^**,**^c^**), mean statistically significant differences between treatments. (*p* < 0.05).

**Table 2 molecules-25-03183-t002:** Comparative table of amount cooper in MIC, MBC and IC_50_
*on F. psychrophilum* of the copper (I) complex [Cu(NN_1_)_2_]ClO_4_ and precursor Cu(I) salt ([Cu(CH_3_CN)_4_]ClO_4_).

Copper (I) Complex [Cu(NN_1_)_2_]ClO_4_			Precursor Salt [Cu(CH_3_CN)_4_]ClO_4_	
	Amount Total (µg)	Amount Cu (µg)	Amount Total (µg)	Amount Cu (µg)
IC_50_	16.1	1.3	10.4	2.0
MIC	32	2.7	64	12.5
MBC	32	2.7	64	12.5

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
