# Peer review of "New Copper(I) Complex with a Coumarin as Ligand with Antibacterial Activity against Flavobacterium psychrophilum"

_molecules, 2020, doi:10.3390/molecules25143183_

Round 1
Reviewer 1 Report
Dear Editor,
I carefully revised the manuscript “molecules-855776” Titled “New copper(I) complex with natural product coumarin as ligand with antibacterial activity against Flavobacterium psychrophilum” by Maialen Aldabaldetrecu, Mick Parra, Sarita Soto, Pablo Arce, Mario Tello, Juan Guerrero, Brenda Modak.
The paper is not clearly presented, and I could detect several errors, which unfortunately seriously undermine the overall judgement on the work. In my opinion, the work is not worthy of publication on Molecules in its present form.
Only a few main observations are given below:
- Line 27: “This complex shows have antibacterial activity…”
- Line 83: The compound Pyridine 2-carboxaldehyde is reported, while in the synthesis scheme (Figure 1) and the experimental section, the 2-quinoline carboxaldehyde is reported. The two compounds have different structures (the 2-quinoline carboxaldehyde has an extra benzene ring and looking at the synthesis scheme, it is the correct compound).
- Line 93 (Fig.1): Error in the name of the compounds: 2-quinolin carboxaldehyde= 2-quinoline carboxaldehyde; Bencene=Benzene).
- Lines 132-134: it seems that the same sentence is repeated twice.
- Lines 142-143: As reported "The synthesized complex shows a high stability to oxidation in different solvents and different temperatures". From what did the Authors deduce this? What analysis do they refer to? If it is already reported in the literature, please quote a reference.
- Lines 182-185: I think the concept of cell permeability and lipophilicity after the coordination of the metal by the ligand is not well explained.
- Lines 187-190: Here again, it is not very clear. The Authors state that they don't report in the work the mechanism of the antibacterial activity of the complex. Still, then they mention something that, in my opinion, is not very clear. Please clarify.
- Line 223: “The experiments were performed in triplicate; however, the results were the same in each replica.” This is something tough to reach.
- I could not find the results of the statistical analysis.
- Line 224: “Different letters mean difference between treatments. Statistical analyzes were performed separately for bacteria and for each cell line.” This sentence is not clear. Where can be different letters found?
- Line 227: ina Table 2, some columns are redundant (Cu amount, % Cu).
- Experimental section: mass spectra are not provided. Please add.
- Line 239: dichloromethane typing error.
- Lines 241-242: several errors: potenciostate, electroquemical analizer, sofware probided suplied by…and many more.
Reviewer 2 Report
This study reports the preparation and characterization of a copper complex with an imine ligand containing a coumarin unit and evaluation of its antibacterial activity against a bacterium responsible for diseases of freshwater fish species.
The topic fits well the journal scope but the manuscript may be significantly improved with regards to presentation of data and comments. In general, data are presented without much comments and a number of paragraphs/sentences are difficult to follow. The language should be checked throughout the text
Introduction: line 48 alpha-pyrene should be 2-pyranone or alpha-pyrone
Lines 68 and ff . Referring to coumarin as natural product is correct but once a significant chemical manipulation has been done the potential advantage (in terms of toxicity or sustainability) is lost. This should be more clearly indicated also in the title. This could be modified as “a new copper complex with a coumarin based ligand with antibacterial…”
Results and discussion: the synthesis is described very briefly with no details on the procedure and the yields. Also in the experimental section no fractionation/purification of the reaction products is described which appears quite surprising.
Scheme 1 : check the legend. Check the name of the quinoline .. check spelling of benzene, hv is not appropriate for a microwave promoted reaction.
UV Vis: description of the UV Vis feature is not clear at all . Lines 132-135 are a repetition. Line 136-143 Homo Lumo please check it is rather obscure perhaps also several clerical mistakes
Cyclic voltammetry: data are clear enough but what are the implications ?
lines 162-165 what does it mean major reactivity? the whole sentence is not clear enough
Antibacterial test: line 191-195 please check and rephrase
Cytoxicity tests. It appears that the complex is rather toxic at low doses although with some differences depending on the cell lines which would restrict the field of possible applications. It seems that coumarin ligand enhances the toxicity of copper which again is not positive. This represents a major limitation and should be highlighted and cautioned more clearly
Reviewer 3 Report
The article is focused on evaluation and synthesis a new antibiotic which can be used to treat Flavobacterium psychrophilum generated large economic losses in the freshwater salmonid aquaculture industry. Due to increasing antibiotic resistans in living organisms looking for the new antibacterial agents is strongly expected. The idea of ​​authors to use combination of coumarins and cooper (I) ions against bacteria can be justified in the light of the available literature. Both factors show a broad spectrum of antibacterial activity based on different mechanisms, so their combination can be more effective. In the article, the authors proposed a method of synthesis a new antibiotic with high efficiency, desribing in detail steps of production: nitration of coumarin, reduction, condensation and Cu (I) complex formation. Next they confirmed stability of new compound in aqueous solution using three different methods NMR, UV-Vis and cyclic voltammetry. The final part of article contains the results of biological activity evaluation of obtained substance. Authors checked the antibacterial power against F. psychrophilum isolated from fish. On the other hand, using three cell lines they control the potential toxicity of proposed antibiotic for treated fish. The promising results showed that the treatment dose is by 2-3 times lower than IC50 concentration for fish embryo, kidney and intestine cells.
In view of the facts cited, I rate the idea and its scientific realization very highly. The work provides valuable information on how to obtain and confirm the effectiveness of a new antibiotic for aquaculture industry, which of course requires in vivo testing before implementation.
In general, the paper is properly written, I have only one important critical remark and a few minor errors:
1.The use of the term "natural coumarin" suggests that a compound of natural origin was used while you used synthetic coumarin derivative. I suggest removing the word "natural" from the title (and its repetitions in text). It is sufficient to clarify in the "Introduction" that coumarin is a plant metabolite, abundantly found in cymnamon for example.
2. line 37 check bacteria name
3. line 59-66: You describe mechanism of antibacterial ation of Cu(I)? Add it.
4. line 202: flow cytometry
5. line 211:growth stage
6. line 226-227:Where did you comment the results presented in Table 2?
7. line 312:check THYES
8. In "References" the bacterial names in original titles were probably written in Italic
Round 2
Reviewer 1 Report
Dear Editor,
the Authors have responded accurately to all the comments made. Regarding English, however, my indications were certainly not exhaustive. The work still needs an extensive English revision before publication.
Reviewer 2 Report
The authors duly addressed most of the criticisms raised. The style of the presentation still needs revision and repetitions (line 144-148 of the revised version) should be checked and removed